# Primary care service utilisation and outcomes in type 2 diabetes: a longitudinal cohort analysis

Sam Hodgson [1], Jeffrey Morgan-Harrisskitt,[2] Hilda Hounkpatin,[1] Beth Stuart,[1] Hajira Dambha-Miller [1]

¹Primary Care Research Centre, University of Southampton, Southampton, UK
²NHS South Central and West Commissioning Support Unit, Newbury, UK

**Correspondence to**
Dr Sam Hodgson;
samcbhodgson@gmail.com

## ABSTRACT

**Objectives** To describe primary care utilisation patterns among adults with type 2 diabetes and to quantify the association between utilisation and long-term health outcomes.

**Design** Retrospective cohort study.

**Setting** 168 primary care practices in Southern England within the Electronic Care and Health Information Analytics database between 2013 and 2020.

**Participants** 110 240 adults with Quality and Outcomes Framework read code of type 2 diabetes diagnosis; age greater than 18 years; linked and continuous records available from April 2013 until April 2020 (or death).

**Primary and secondary outcome measures** (1) Rates of service utilisation (total number of primary care contacts per quarter) across the study period; (2) participant characteristics associated with higher and lower rates of service utilisation; and (3) associations between service utilisation and (A) cardiovascular disease (CVD events) and (B) all-cause mortality.

**Results** Mean (SD) number of primary care attendances per quarter in the cohort of 110 240 went from 2.49 (2.01) in 2013 to 2.78 (2.06) in 2020. Patients in the highest usage tertile were more likely to be female, older, more frail, white, from the least deprived quintile and to have five or more comorbidities. In adjusted models, higher rates of service utilisation (per consultation) were associated with higher rates of CVD events (OR 1.0058; 95% CI 1.0053 to 1.0062; p<0.001) and mortality (OR 1.0057; 95% CI 1.0051 to 1.0064; p<0.001).

**Conclusions** People with type 2 diabetes are using primary care services more frequently, but increased volume of clinical care does not correlate with better outcomes, although this finding may be driven by more unwell patients contacting services more frequently. Further research on the nature and content of contacts is required to understand how to tailor services to deliver effective care to those at greatest risk of complications.

### Strengths and limitations of this study

► This study includes a large sample of 110 240 individuals with a relatively long follow-up period of 7 years.

► This study analyses robust and precise measures of clinical parameters from real-world clinical data.

► The retrospective and observational nature of the study design limits permissible inferences on causality or directionality of associations.

► Our data do not capture information on the nature, quality or duration of primary care contacts nor the relevance of that contact to type 2 diabetes.

► Although the study population is large, it is taken from an area of the UK (Southern England) that is more affluent, less ethnically diverse and less socio-economically deprived than the UK average, which may limit generalisability of findings.

NHS long-term plan builds on the General Practice Forward View articulating the need for better prevention of cardiovascular disease (CVD).[3] One key area of focus is the prevention of type 2 diabetes and its related CVD complications in primary care, where over 80% of the disease is managed.[4] It is hypothesised that primary care contacts could provide opportunities to positively influence and support healthy behaviours, optimise treatment plans, promote self-management and detect ill-health earlier.

For this reason, the last few years have seen greater provision of primary care diabetes services through increased multidisciplinary teams, automated recall and greater efforts at monitoring of diabetes progress through the UK National Diabetes Audit,[5] a national surveillance programme assessing the quality of type 2 diabetes care in UK primary care, and Quality Outcome Framework (QOF).[6] However, both the QOF and National Diabetes Audit have demonstrated large variations in primary care diabetes uptake, while long-term outcomes remain poor.[7 8] Type 2 diabetes is still the leading cause of

## INTRODUCTION

Diabetes UK estimate that by 2030 5.5 million individuals in the UK will be diagnosed with type 2 diabetes,[1] which today costs the National Health Service (NHS) over £10 billion a year.[2] Strategies to optimise treatment and prevent associated complications are therefore a public health priority. The UK Government's

premature mortality and for someone diagnosed at the age of 50 years, for example, average life expectancy in the UK is reduced by 6 years.[9] Despite increased research, funding and the development of a number of interventions to prevent diabetes complications, few have been effectively translated in primary care.[10–13] In part, this may reflect the fact that national strategies and previous diabetes research do not adequately consider the contextual or real-world challenges in primary care. To develop effective strategies that are relevant and applicable to this setting, it is necessary to first understand the current utilisation and availability of primary care services to understand the context in which any future interventions would be nested.[14]

These contextual issues are critical to examine at a time when primary care is rapidly adapting to increasing demands on services, a growing burden of chronic disease and challenges in workforce recruitment. Cuts to public health and social funding with no planned increases in capital, education, training and local authority public health or social care are likely to add to existing pressures.[15–18] Several studies have already described patterns of service utilisation, characteristics of service users and how these relate to health outcomes in general terms, but these have not specifically considered type 2 diabetes in primary care,[19] nor long-term complications such as heart attacks and strokes that have the greatest impact of patient's lives and on healthcare costs.[20 21]

The availability of longitudinal electronic primary care health records that include large cohorts with objective measures provides an opportunity to overcome the limitations of previous research in this area and inform future provision. In this study, we examined patterns of primary care service utilisation in adults with type 2 diabetes in a large population-based cohort and quantified the association between utilisation and long-term health outcomes.

## METHOD

### Design

Retrospective cohort analysis.

### Data source

The Electronic Care and Health Information Analytics (CHIA) database includes individual level anonymised live data from primary care records linked to local acute hospital trusts in the South of England (Hampshire). Data from 1.5 million consenting patient medical records have been anonymised and collected continuously across 168 primary care practices covering urban and rural populations. Patients automatically 'opt in' to their anonymised data being included in the research data set and can opt out via their primary care provider or the CHIE platform. Data on the proportion of patients opting out was not available to the research team. Data include coded clinical entries on demographic data, service utilisation, diagnoses, investigations, medications and clinical outcomes from primary care and hospital laboratories.

### Study population

Individuals were included if they met the following criteria: age greater than 18 years; linked and continuous records available from April 2013 to April 2020 (or death); and QOF read code of type 2 diabetes diagnosis (including codes C10F.00, C10F.11 and complication-related derivatives; full list of included codes given in online supplemental file 1).

### Exposure

Primary care service utilisation was defined as the total number of contacts during the in-hours service over the study period, which was between April 2013 and April 2020. This included telephone, in-person or home visits recorded in the electronic primary care record.

Outcomes include: (1) CVD events, defined as a composite of myocardial infarction, amputation and stroke and (2) all-cause mortality defined as death from any cause during the study period

### Covariates

#### Baseline sociodemographic data

We extracted data on age, sex and ethnicity. Socioeconomic deprivation was determined by the index of multiple deprivation (IMD), a score calculated by the government to reflect deprivation in a specific geographic area determined based on seven domains of deprivation including income, employment, education, health and crime.[22]

#### Baseline clinical variables

Baseline comorbidities were defined from diagnostic codes of existing QOF conditions and included coronary heart disease, chronic kidney disease, chronic obstructive pulmonary disease (COPD), asthma, cancer, dementia, atrial fibrillation, epilepsy, heart failure, stroke, peripheral vascular disease, hypertension, osteoporosis, osteoarthritis, depression and frailty. Frailty was defined using the Electronic Frailty Index Score, a validated tool used to estimate an individual's frailty state using data automatically held in primary care health records.[23] Latest smoking status (current, ex or never-smoked), weight, body mass index (BMI), systolic and diastolic blood pressure and biochemistry measures (including glycated haemoglobin (HbA1c) total cholesterol, high densirty lipoprotein (HDL) DL-cholesterol and estimated glomerular filtration rate (eGFR)) were examined as the earliest available recordings from 1 April 2013

#### Duration of type 2 diabetes

We estimated the interval between the first code for type 2 diabetes in the electronic record until study entry (April 2013).

### Statistical analysis

Baseline characteristics were summarised using descriptive statistics in table 1. Time series graphs were plotted using mean utilisation for each quarter over the 7-year follow-up. We compared these between sociodemographic

**Table 1** Summary sociodemographic and clinical characteristics of the Care and Health Information Analytics type 2 diabetes cohort, comprising all patients diagnosed with type 2 diabetes in a linked primary and secondary care electronic database in Hampshire, UK (n=110 240)

| Sex, n (%) | | Patients missing data, n (%) |
|---|---|---|
| Female | 48 706 (44.18) | 0 (0) |
| Male | 61 534 (55.82) | |
| Age (years) | | |
| Age | 62.89 (14.29) | 0 (0) |
| Ethnicity, n (%) | | |
| Asian | 3295 (2.99) | 54 011 (48.99) |
| Black | 498 (0.45) | |
| Mixed or other | 810 (0.73) | |
| White | 51 626 (46.83) | |
| Index of multiple deprivation, n (%) | | |
| 1 | 14 035 (12.73) | 1075 (0.98) |
| 2 | 21 833 (19.8) | |
| 3 | 21 064 (19.11) | |
| 4 | 23 062 (20.92) | |
| 5 | 29 171 (26.46) | |
| Smoking status, n (%) | | |
| Current smoker | 13 875 (12.59) | 492 (0.45) |
| Ex smoker | 44 875 (40.71) | |
| Never smoked | 50 998 (46.26) | |
| Total comorbidities | | |
| Total comorbidities | 2.36 (1.79) | 0 (0) |
| Individual comorbidities, n (%) | | |
| Stroke | 7694 (6.98) | 0 (0) |
| Myocardial infarction | 9812 (8.9) | 0 (0) |
| Peripheral vascular disease | 3565 (3.23) | 0 (0) |
| Chronic kidney disease | 15 471 (14.03) | 0 (0) |
| Dementia | 3558 (3.23) | 0 (0) |
| Total medications prescribed in study period | 48.59 (30.12) | 0 (0) |
| Frailty score | 0.22 (0.11) | 77 (0.07) |

Values are presented as mean (SD) unless otherwise stated; all values are taken at baseline.

variables. Service utilisation was summarised with descriptive statistics and divided into tertiles in line with previous approaches in the literature[24] in table 2; we tested for differences between tertiles using one-way analysis of variance for continuous variables and a $\chi^2$ test for categorical variables. Finally, we constructed univariate and multivariable logistic models on a complete-case analysis to examine the association between service utilisation (total number of contacts over the study duration), CVD events and all-cause mortality.

Multivariable logistic regression models were adjusted for potential confounders on a priori reasoning including baseline sociodemographic variables (age, sex, ethnicity and IMD); clinical variables to take into account disease severity as sicker patients might attend more frequently (HbA1c, blood pressure, total cholesterol, BMI, frailty, total number of medications prescribed across the study period and smoking status); and practice level covariates (to account for clustering). Parameter estimates are presented with 95% CIs. Tests were conducted as two sided at the 5% significance level. Data were analysed using STATA V.16.1 and R V.4.0.3. Our findings are reported in line with the Strengthening the Reporting of Observational Studies in Epidemiology and RECORD guidelines for observational studies using routinely collected health data.[25]

**Table 2** Sociodemographic and clinical characteristics of the Care and Health Information Analytics (CHIA) type 2 diabetes cohort, presented by primary care utilisation tertiles, comprising all patients diagnosed with type 2 diabetes in a linked primary and secondary care electronic database in Hampshire, UK (n=75 701)

| Variables | All participants | Tertile 1 (low usage) | Tertile 2 | Tertile 3 (high usage) | P value |
|---|---|---|---|---|---|
| Complete case cohort | | | | | |
| Number of patients (n) | 75 701 (0) | 25 234 (0) | 25 234 (0) | 25 233 (0) | |
| Average attendances, mean (SD) | 74.04 (44) | 38.6 (8.49) | 63.8 (7.87) | 119.73 (47.17) | <0.01 |
| Sex, n (%) | | | | | |
| Female | 34 887 (46.09) | 9959 (39.47) | 11 739 (46.52) | 13 189 (52.27) | <0.01 |
| Male | 40 814 (53.91) | 15 275 (60.53) | 13 495 (53.48) | 12 044 (47.73) | <0.01 |
| Age at start (years) | | | | | |
| Age, mean (SD) | 63.15 (13.04) | 60.2 (12.78) | 63.05 (12.77) | 66.19 (12.87) | <0.01 |
| Age group at start (years), n (%) | | | | | |
| 18–44 | 6506 (8.59) | 2770 (10.98) | 2102 (8.33) | 1634 (6.48) | <0.01 |
| 45–64 | 31 054 (41.02) | 12 436 (49.28) | 10 491 (41.57) | 8127 (32.21) | <0.01 |
| 65–84 | 35 804 (47.3) | 9510 (37.69) | 11 931 (47.28) | 14 363 (56.92) | <0.01 |
| 85+ | 2337 (3.09) | 518 (2.05) | 710 (2.81) | 1109 (4.4) | <0.01 |
| Ethnicity, n (%) | | | | | |
| Asian | 2097 (2.77) | 730 (2.89) | 841 (3.33) | 526 (2.08) | <0.01 |
| Black | 309 (0.41) | 110 (0.44) | 118 (0.47) | 81 (0.32) | 0.03 |
| Mixed or other | 544 (0.72) | 195 (0.77) | 182 (0.72) | 167 (0.66) | 0.34 |
| White | 39 519 (52.2) | 11 272 (44.67) | 13 543 (53.67) | 14 704 (58.27) | <0.01 |
| Missing data | 33 232 (43.9) | 12 927 (51.23) | 10 550 (41.81) | 9755 (38.66) | <0.01 |
| Index of multiple deprivation quintile, n (%) | | | | | |
| 1 | 9301 (12.29) | 3311 (13.12) | 3013 (11.94) | 2977 (11.8) | <0.01 |
| 2 | 14 215 (18.78) | 4857 (19.25) | 4621 (18.31) | 4737 (18.77) | 0.01 |
| 3 | 13 924 (18.39) | 4889 (19.37) | 4477 (17.74) | 4558 (18.06) | 0.56 |
| 4 | 16 285 (21.51) | 5415 (21.46) | 5560 (22.03) | 5310 (21.04) | 0.01 |
| 5 n(%) | 21 368 (28.23) | 6612 (26.2) | 7348 (29.12) | 7408 (29.36) | 0.03 |
| Missing IMD values, n (%) | 608 (0.8) | 150 (0.59) | 215 (0.85) | 243 (0.96) | <0.01 |
| HbA1c (%) | | | | | |
| HbA1c, mean (SD) | 7.55 (1.53) | 7.43 (1.48) | 7.51 (1.49) | 7.65 (1.59) | <0.01 |
| HbA1c (mmol/mol) | | | | | |
| HbA1c, mean (SD) | 59.1 (16.97) | 57.85 (16.38) | 58.73 (16.49) | 60.21 (17.65) | <0.01 |
| BMI (kg)/m$^2$ | | | | | |
| BMI, mean (SD) | 31.9 (7.31) | 31.2 (7.21) | 31.1 (7.19) | 32.2 (7.38) | <0.01 |
| Electronic frailty index score | | | | | |
| Mean electronic frailty index score (SD) | 0.22 (0.10) | 0.16 (0.08) | 0.22 (0.09) | 0.30 (0.11) | <0.001 |
| Number of comorbidities, n (%) | | | | | |
| ≤2 | 40 400 (53.37) | 18 048 (71.52) | 13 659 (54.13) | 8693 (34.45) | <0.01 |
| 3–4 | 23 970 (31.66) | 5996 (23.76) | 8575 (33.98) | 9399 (37.25) | <0.01 |
| ≥5 | 11 331 (14.97) | 1190 (4.72) | 3000 (11.89) | 7141 (28.3) | <0.01 |
| Total number of medications prescribed, n (%) | | | | | |
| ≤2 | 1078 (1.42) | 737 (2.92) | 235 (0.93) | 106 (0.42) | <0.01 |

Continued

**Table 2** Continued

| Variables | All participants | Tertile 1 (low usage) | Tertile 2 | Tertile 3 (high usage) | P value |
|---|---|---|---|---|---|
| 3–5 | 1037 (1.37) | 674 (2.67) | 285 (1.13) | 78 (0.31) | <0.01 |
| 6–9 | 1498 (1.98) | 893 (3.54) | 419 (1.66) | 186 (0.74) | <0.01 |
| ≥10 | 72 088 (95.23) | 22 930 (90.87) | 24 295 (96.28) | 24 863 (98.53) | <0.01 |

Data sourced from CHIA dataset values are presented as mean (SD) unless otherwise stated; all values are taken at baseline.
BMI, body mass index; HbA1c, glycated haemoglobin; IMD, index of multiple deprivation.

## Patient and public involvement

Patients and the public were not directly involved in this research. The nature of the anonymised records means individual participants could not be involved.

## RESULTS
### Participant characteristics

The cohort included 110 240 people with type 2 diabetes across 168 primary care practices in Southern England. The mean (SD) age of participants at baseline was 62.9 (14.3) years, there were more men than women (55.8%), and over a quarter of the cohort came from the highest socioeconomic quintile (IMD 5, 26.5%). The characteristics of participants within the cohort are summarised in table 1. We compared participants with and without missing data, with ethnicity having the largest number of missing data (48.2%), which is not unusual for routinely collected primary care data.[26] The majority of participants with a coded ethnicity were white (90.1%). Participants with missing data were more likely to be male and from the highest socioeconomic quintile.

### Primary care service utilisation

Across the whole study period (April 2013–April 2020), mean number of attendances per patient was 74.0 (2.68 per quarter, SD 2.04). Seasonal variations were observed within each year (with higher use in winter), but overall mean (annual rolling average) utilisation for the cohort increased over the study period from 2.48 (2.01) per quarter in 2013 to 2.78 (2.06) attendances per quarter in 2020 (figure 1). This increase was observed across all age groups (figure 2A) and was highest among those aged 65–84 years. Attendance rates were higher in white patients compared with black, Asian or other ethnic minority backgrounds (figure 2B).

### Characteristics of patients by service utilisation tertile

Sociodemographic and clinical characteristics of included individuals are described by service utilisation tertile in table 2. Mean (SD) utilisation of primary care services was four times greater in the highest utilisation tertile (120 (47.1) attendances) compared with the lowest (38.6 (8.9) attendances) (p<0.001). Patients in the highest usage tertile were more likely to be female, older, more frail, white, from the least deprived IMD quintile and to have five or more comorbidities. Patients in the lowest usage

tertile were more likely to be male, younger, non-white, less multimorbid, less frail and from the most deprived IMD quintile. Mean (SD) HbA1c was higher in the highest utilisation tertile (60.2 (17.65) mmol/mol)) than the lowest utilisation tertile (57.9 (16.8) mmol/mol).

### Primary care utilisation and outcomes

In both univariate and maximally adjusted models (table 3), higher rates of service utilisation were associated with higher rates of CVD (adjusted OR 1.0057 per additional primary care attendance; 95% CI 1.0052 to 1.0061; p<0.001) and mortality (adjusted OR 1.0056; 95% CI 1.0050 to 1.0063; p<0.001).

## DISCUSSION
### Summary of key findings

In this retrospective cohort study of 110 240 people with type 2 diabetes, we observed growing rates of primary

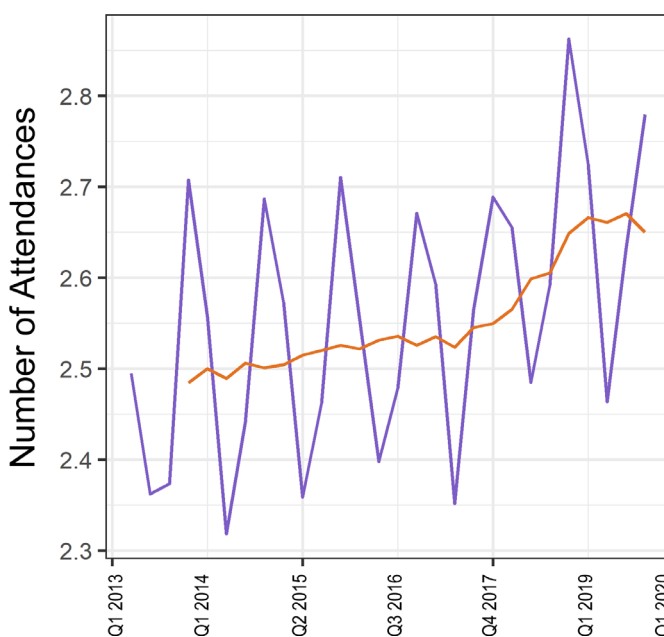

**Figure 1** Mean number of primary care attendances by year quarter (Q1–Q4) between 2013 and 2020 in the CHIA type 2 diabetes cohort. Q1 is defined as January–March. CHIA, Care and Health Information Analytics.

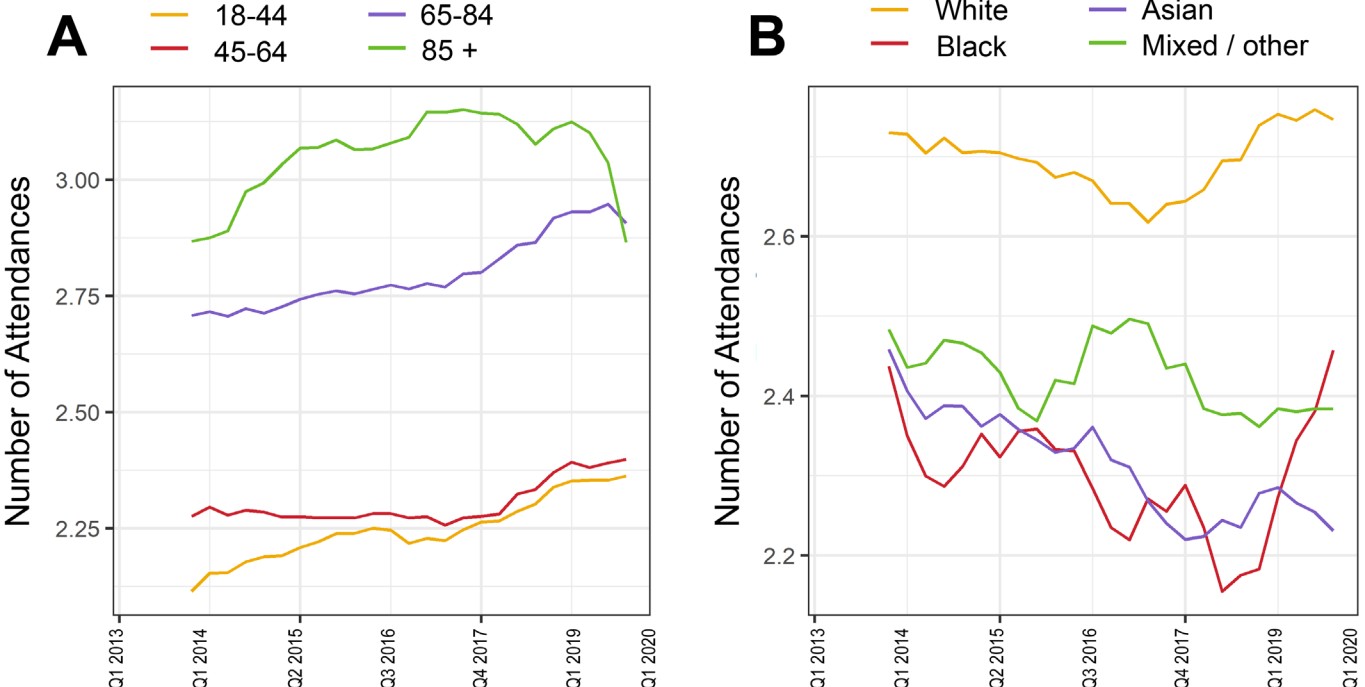

**Figure 2** Rolling four-quarter mean number of primary care attendances by year quarter between 2013 and 2020 in the CHIA type 2 diabetes cohort stratified by (A) age band (in years) at study entry and (B) ethnicity. Q1 is defined as January–March. For example, the mean number of attendances for Q1 2014 represents mean attendance between Q1 2013 and Q1 2014. CHIA, Care and Health Information Analytics.

care service utilisation between 2013 and 2020. This was highest among white females with multimorbidity and polypharmacy from higher socioeconomic backgrounds. After adjusting for potential confounders including age and comorbidities, we find more frequent contacts with primary care is associated with higher risk of CVD events and mortality. These findings could suggest a need to focus on personalisation of care, and quality of delivered care, to improve outcomes in type 2 diabetes or could reflect reverse causality, that is, services are being accessed more frequently by those with greater health needs.

### Comparison to existing literature
Our findings are in agreement with those of Hobbs *et al* describing trends of increasing primary care utilisation between 2007 and 2014 in England.[19] While most studies in this domain have examined trends in general utilisation of primary care, our study is novel in specifically examining patters of service use among individuals with type 2 diabetes.

In a recent cross-sectional observational study assessing routine primary care data from 2.7 million English patients, Lay-Flurrie *et al*[27] report no association between service utilisation and mortality. In their analysis, the authors did not stratify by underlying diagnosis, and just 16.5% of included patients were aged over 65 years, in contrast to 50.4% in our cohort. Taken in tandem with our findings that increased service utilisation is associated with mortality in adults with type 2 diabetes, these findings imply service utilisation might offer benefit as a predictor of poor outcomes in selected primary care populations, such as older adults or those with type 2 diabetes

**Table 3** Univariate (unadjusted) and multivariate (adjusted) models describing the association between primary care service utilisation, defined as number of contacts with primary care services, cardiovascular disease and mortality, in the CHIA data set between 2013 and 2020

| | Unadjusted OR (95% CI) per additional primary care contact | Univariate p value | Adjusted* OR (95% CI) per additional primary care contact | Multivariate p value |
|---|---|---|---|---|
| Cardiovascular disease† | 1.0071 (1.0066 to 1.0074) | <0.001 | 1.0057 (1.0052 to 1.0061) | <0.001 |
| Mortality | 1.0077 (1.0070 to 1.0082) | <0.001 | 1.0056 (1.0050 to 1.0063) | <0.001 |

*Adjusted for age, sex, ethnicity, index of multiple deprivation, smoking status, number of prescribed medications, frailty, BMI, HbA1c, systolic and diastolic blood pressure, mean total cholesterol and practice-level comorbidity prevalence to account for local clustering.
†Defined as a composite of myocardial infarction, amputation and stroke.
BMI, body mass index; CHIA, Care and Health Information Analytics; HbA1c, glycated haemoglobin.

We find that in patients with type 2 diabetes higher numbers of primary care attendances were associated with a higher odds of CVD and mortality, even after taking into account comorbidities and sociodemographic factors. One explanation for this finding could be that optimal preventive care cannot always be achieved in the context of services poor in time and funding resources, despite high frequency contacts. This would be supported by findings from semistructured interviews in patients with type 2 diabetes, General Practitioners (GPs) and specialist nurses, which find that only 'minimum care standards can be maintained' in providing care to patients with type 2 diabetes.[28]

Our study does not inform on the mechanisms underlying the associations between service utilisation, CVD and mortality. It is possible reverse causality[29] could underlie this finding: patients with established CVD, for example, are more likely to use primary care services for medication reviews and routine care and also more likely to develop further CVD or die.[30] Patients who are more frail and multimorbid, or with more difficult to manage or refractory disease, may also be more likely to use primary care services more frequently,[31] or to be followed up more intensively by their clinicians, although we adjusted for these potential confounders in our analysis. Alternatively, this finding could reflect underlying mechanisms not captured in our adjusted models, such as a higher burden of non-CVDs associated with mortality such as cancer and COPD.[32]

### Strengths and limitations

Strengths of our study include its large sample of 110 240 people and a relatively long follow-up period of 7 years with robust and precise measures of clinical parameters and restriction of analysis to complete cases. Limitations include the retrospective, observational nature of our study design that limits permissible inferences on causality or directionality of associations; it is unclear from our findings why a relationship between higher rates of service utilisation and poor outcomes exists in type 2 diabetes, and reverse causality may well underlie these findings, that is, more unwell patients requiring more frequent care and being invited for more regular reviews. Our data do not capture information on the nature, quality or duration of primary care contacts; who the contact was with (eg a GP or specialist nurse); nor the relevance of that contact to type 2 diabetes. For example, patients attending once yearly for a 30 min review of their type 2 diabetes would appear to be using services to a lesser extent than patients attending three times per year for 10 min appointments with minor ailments but may be receiving more focused, high-quality diabetes care. Although the study population is large and includes a large geographic area in the UK, it does include a more affluent, less ethnically diverse and less socioeconomically deprived population than the UK average, which may limit generalisability of findings. Relatively high rates of missing data on participant ethnicity are in keeping with typical results taken from real-world primary care databases, but these missing data are an important limitation to interpretation of findings related to ethnicity, and capture of these data in future observational cohorts would be of clear benefit. Furthermore, we did not further substratify patient groups by, for example, HbA1c levels or comorbid conditions such as hypertension. This approach could identify particularly high-risk groups and could form the basis for future research in this field.

### Implications for research and practice

This study shows an increase of 8.3% in rates of primary care service utilisation by patients with type 2 diabetes between 2013 and 2020. Highest rates of utilisation were seen in older and frailer patients. High levels of missing data regarding patient ethnicity may limit understanding, and future focus on capturing this within electronic health records may be of benefit in redressing health inequalities.

Our finding that increased service utilisation is not associated with better outcomes in type 2 diabetes suggests simple expansion of number of primary care contacts may not necessarily be beneficial and that there is a need to consider the quality, nature and content of contacts when tailoring service design in type 2 diabetes. Further research could highlight knowledge gaps in understanding the frequency and content of cardiovascular risk assessments, lifestyle and medication reviews in high-risk individuals, for example.

As opposed to achieving better disease control and developing fewer complications, patients with more frequent primary care contacts had higher HbA1c, higher rates of CVD and higher rates of mortality. Although these patients tended to be older and more multimorbid, these associations persisted after adjustment for these and other confounders. Further research capturing the nature, content and duration of contacts in relation to delivery of care for type 2 diabetes is needed to understand the mechanisms underlying this association and address contributory factors.

**Contributors** HD-M designed the study, wrote the first draft of the paper, edited and contributed to subsequent versions. SH analysed the data and drafted and revised subsequent versions of the paper. JM-H carried out data analysis and revised the paper. HH revised the paper. BS provided advice on statistical methods and revised the paper. HDM is guarantor.

**Funding** HD-M is a National Institute for Health Research (NIHR) funded Academic Clinical Lecturer and has received NIHR School of Primary Care Research funding to support this work (SPCR2014-10043). SH is an Academic Clinical Fellow funded by the NIHR.

**Disclaimer** The views and opinions expressed by authors in this publication are those of the authors and do not necessarily reflect those of the UK NIHR or the Department of Health and Social Care.

**Competing interests** None declared.

**Patient consent for publication** Not applicable.

**Ethics approval** The Hampshire Health Record (Care and Health Information Exchange) has its own independent panel for ethics and governance approval. We have been granted governance and ethics approval. We have additionally received

approval from the University of Southampton Research and Governance office (submission reference: ERG049232).

**Provenance and peer review** Not commissioned; externally peer reviewed.

**Data availability statement** Data may be obtained from a third party and are not publicly available. Access to the dataset used for this research is governed by the Hampshire Health Record (Care and Health Information Exchange), which has its own independent panel for ethics and governance approval and data access. More information on their procedures can be found here: https://careandhealthinformationexchange.org.uk/wp-content/uploads/2018/04/CHIE-Review-of-Procedures-Compliance-with-GDPR_V5-1.pdf.

**ORCID iDs**
Sam Hodgson http://orcid.org/0000-0002-5610-850X
Hajira Dambha-Miller http://orcid.org/0000-0003-0175-443X

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
