## [Reviewer comments · BMJ Open]

ARTICLE DETAILS

TITLE (PROVISIONAL)	Primary care service utilisation and outcomes in type 2 diabetes: A longitudinal cohort analysis
AUTHORS	Hodgson, Sam; Morgan-Harrisskitt, Jeffrey; Hounkpatin, Hilda; Stuart, Beth; Dambha-Miller, Hajira

VERSION 1 – REVIEW

REVIEWER	Daly, B. University of Auckland
REVIEW RETURNED	20-Aug-2021

GENERAL COMMENTS	20/8/2021 Review for BMJ Open 'ID bmjopen-2021-054654' Titled 'Primary care service utilisation and outcomes in type 2 diabetes: A longitudinal cohort analysis' General feedback This retrospective large cohort study reports on trends in general practice utilisation and patient characteristics and their association with major cardiovascular (CV) events and mortality. A total of 110,240 people with type 2 diabetes were followed in Southern England between 2013 and 2020. Appropriate statistical tests have been used for continuous and categorical variables while controlling for confounding variables in the multivariate analysis. However, numbers of patients with data and included in the analysis need to be clearly outlined in the methodology. The discussion needs restructuring and comparison with large international reports. The authors need to adhere to the usual academic essay style of writing and correct grammatical errors – outlined below. The STROBE check list does not appear to be completed. Abstract
---

This is generally well written but check the format as it doesn't appear to comply with BMJ open. Avoid abbreviations in the abstract.

Introduction

Well written and contextualises this study. Please see grammatical and reference comments below.

Methodology

Please explain how consent was obtained and are patients asked to opt out i.e. what is the default position. What proportion of the total patients registered consented or did not consent for representation / generalisability. Appropriate statistical tests have been used for continuous and categorical variable and controlling for available confounding variables in the multivariate analysis.

Results

Put results for ethnicity in context. I suggest you explain that ethnicity is only available for 51% and then present proportions for those people. This contextualises ethnicity within the total population i.e. approximately 85% are English, Irish and Welsh. Note a 'majority' is at least >50%.
Round off data in the text to whole numbers unless very small. Re comment about missing data – how do you know these patients were mostly 'white' or are you assuming that from their IMD?
Please explain differences between table 1 and 2 in the text and in your methodology – this is not clear.

Table 1

Include numbers in the title (remove from table) and use a better description of CHIA – avoid abbreviations in the title.
Review lay out – consider double columns to shorten and widen.
Move SD beside mean.
Proportions should be for those with data especially ethnicity due to a large proportion of missing data. Add this to the main limitations and recommend that this data is captured from all patients going forward as this is important for equity issues.
Last cell contents should be in a footnote as only for age and medications. Please check the latter i.e. ? 49 medications per person.

Table 2

Numbers are not clear. The total should be in the title.
The title should clearly differentiate these analyses from table 1. You need an in-text reference.
Ethnic proportions should not include missing data – see above.
Check spacing.

Consider commas in numbers as they are hard to read when so large.

Table 3

Review layout of Table 3

Shorten heading and avoid repetition with headings in the table. The description in title is a repetition of your methods so shorten and reword this.

Use a footnote for the CV events definition.

Review footnote - the OR is the header row so you don't need to define that.

For the footnote use symbols not 'model' for confounding variables.

Review spacing and fonts.

Figures 1 and 2

Suggest you add Q1 – Q4 in titles i.e. ...by year quarters (Q1 – Q4). Description of quarters is in your text.

Centre titles and A and B are usually written as 2a and 2b – check Journal format.

Discussion

Some restructuring is required. Separate paragraphs for increased utilisation and non-associations with improved outcomes and link these comments to other literature.

Move strengths and limitations to the end.

Page 6 lines 45-49 – suggest this as your conclusion.

Page 6 lines 55-57 - this is not clear. Are you referring to your results and state for which patients might benefit rather than 'selected' and 'unselected'.

Lines 59-60 is a repetition of your first paragraph – I suggest you link this into comparative studies.

Page 7 lines 3-8 are not very clear and you are citing qualitative self-reporting from GPs and specialist nurses on the constraints and the effect they might have on patients at high risk of major CV events. What quantitative reports could you cite. This association between primary care consultations and outcomes has been identified in other countries and highlights the many challenges and limitations in managing and supporting people with type 2 diabetes and particularly those already at high risk of related complications including major CV events.

Your findings suggest that increasing consultations is not reducing major events and mortality. Do you have data on the proportion of patients with high HbA1c, hypertension, smoking and high LDL-C and prescription medications? What proportion of patients require further pharmaceutical interventions? You may be planning another paper with this information, but you could indicate this or recommend further analysis to identify gaps in CV risk management.

How do mean numbers of consultations compare with international reports for type 2 diabetes and major outcomes.

	Implications – first para is another summary of results. Can you comment on what is required to identify gaps in practice to reduce the risk profile in those at high risk of CV events? How often do these high risk patients have a CV risk assessment and review of medications and lifestyle? Grammar Abbreviations – check throughout i.e. UK, NHS, IMD. Either use CV or CVD and check as sometimes you have CVD then disease written afterwards. Use correct in-text references style and spacing for this Journal. Check and reference all data i.e. 80% in first para P. 4 lines 13-14 – check punctuation. Check Journals use of small headings. I strongly suggest you reduce the number of small headings and use full sentences and paragraph structure especially noticeable in the methods section. UK spelling needs to be consistent. Capitals when referring to Tables and Figures in the text. Check UK census and follow this where possible for ethnic groups i.e. 'white' is not the same as English, Irish and Welsh. P 5 lines 53-60 check parenthesis, spacing and sentence structure for mean and SD. Review journal lay out of tables and titles. Consider commas in large numbers for readability. Page 5 line 47-49 join sentences re age and ethnicity. 1
--	--

REVIEWER	Trout, Kimberly University of Pennsylvania School of Nursing, Family and Community Health
REVIEW RETURNED	14-Oct-2021

GENERAL COMMENTS	This study is a longitudinal retrospective cohort study that explores the number of primary care visits for adults with type 2 diabetes during the time period April 2013 through April 2020. The findings demonstrate that higher utilization of primary care services resulted in a higher number of cardiovascular events and all-cause mortality. In Table 2, mean weight is presented as a variable, but it would be much more informative if the authors could include mean BMI and adjust findings for BMI. The discussion clearly suggests the limitations of the study (including the retrospective, observational nature of the study design) therefore, limiting inferences on causality and also, the limited racial/ethnic diversity of the population studied. While the authors acknowledge that their findings may be attributed to "reverse causality" (i.e., sicker patients needing more frequent care), they also suggest that their findings demonstrate that increasing frequency of services does not necessarily translate into improved outcomes. The call to conduct more extensive research on the quality of visits, rather than simple quantity is justified.
---

VERSION 1 – AUTHOR RESPONSE

Reviewer 1

General feedback

This retrospective large cohort study reports on trends in general practice utilisation and patient characteristics and their association with major cardiovascular (CV) events and mortality. A total of 110,240 people with type 2 diabetes were followed in Southern England between 2013 and 2020. Appropriate statistical tests have been used for continuous and categorical variables while controlling for confounding variables in the multivariate analysis. However, numbers of patients with data and included in the analysis need to be clearly outlined in the methodology. The discussion needs restructuring and comparison with large international reports. The authors need to adhere to the usual academic essay style of writing and correct grammatical errors – outlined below.

We thank the reviewer for this valuable oversight, and address each point in turn below.

The STROBE check list does not appear to be completed.

Please accept our apologies for inclusion of this draft version of the STROBE checklist in which highlighting alone was used to indicate completion of each STROBE element. An updated checklist has been attached, including page numbers.

Abstract

This is generally well written but check the format as it doesn't appear to comply with BMJ open.

Avoid abbreviations in the abstract.

Under "abstract guidelines" on https://bmjopen.bmj.com/pages/authors/#submission_guidelines the recommended format is "objectives / design / settings / participants / interventions (to be deleted if none in study) / primary and secondary outcome measures / results / conclusions". On reviewing our abstract it appears to us the format does comply with BMJ Open. We are of course happy to amend the format if this is incorrect. Furthermore, we do not believe we include any abbreviations we do not initially define in the abstract (eg cardiovascular disease, CVD) other than "SD", which we have changed to "standard deviation." We have deleted the "QOF" abbreviation as this is not subsequently used.

Introduction

Well written and contextualises this study. Please see grammatical and reference comments below.

No specific changes made on the basis of this recommendation – please see below.

Methodology

Please explain how consent was obtained and are patients asked to opt out i.e. what is the default position. What proportion of the total patients registered consented or did not consent for representation / generalisability.

An additional section has been added to the methods highlighting that patients within the geographical catchment area of the dataset are by default "opted in." Data on the proportion of patients opting out is not available via the CHIE platform or as part of our dataset. We have contacted CHIE to request this information but have not yet received a response prior to the manuscript resubmission deadline.

Appropriate statistical tests have been used for continuous and categorical variable and controlling for available confounding variables in the multivariate analysis.

Results

- Put results for ethnicity in context. I suggest you explain that ethnicity is only available for 51% and then present proportions for those people. This contextualises ethnicity within the total population i.e. approximately 85% are English, Irish and Welsh. Note a 'majority' is at least >50%.

We thank the reviewer for this helpful comment and have contextualised results accordingly, including rephrasing the way in which ethnicity data are presented (ie 46% of participants were coded as white, but 51% of participants had no ethnicity recorded; of those with a recorded ethnicity, 90% were white.).

- Round off data in the text to whole numbers unless very small. Re comment about missing data – how do you know these patients were mostly 'white' or are you assuming that from their IMD?
We have rounded data to 3 significant figures throughout, although we would be happy for rounding of numbers to the nearest integer if the editor feels this aids readability without impacting precision.
We thank the reviewer for this valuable point on description of participants with missing data. The intended meaning was “among participants with a coded ethnicity but other missing data, most were white” – however, we acknowledge this is a contentious means of assessing the characteristics of those with missing data, and have removed this section of the text accordingly.

- Please explain differences between table 1 and 2 in the text and in your methodology – this is not clear.

Table 1 presents a summary of participants in the data set overall. Table 2 reports differences between service utilisation tertiles of these participants. We have adjusted the text to make this distinction clearer in the methods section, and the results section: for table 1 (“The characteristics of participants within the cohortse results are summarised in Table 1.”) and Table 2 (“Sociodemographic and clinical characteristics of included individuals are described by service utilisation tertile in table 2”).

Table 1

- Include numbers in the title (remove from table) and use a better description of CHIA – avoid abbreviations in the title.

This has been adjusted. We are not entirely sure what the reviewer means by “include numbers in the title and remove from table” – presumably they are referring to the total included n? This has been adjusted, but can be changed if required.

- Review lay out – consider double columns to shorten and widen. Move SD beside mean.

We thank the reviewer for this valuable suggestion and have adjusted accordingly by creating a “% missing data” column for each variable.

- Proportions should be for those with data especially ethnicity due to a large proportion of missing data. Add this to the main limitations and recommend that this data is captured from all patients going forward as this is important for equity issues.

We felt it was more representative to highlight precisely which data was missing; presenting only those proportions for non-missing data would not allow the reader to appreciate the likely representativeness of the CHIA cohort and generalisability to their own clinical practice of findings. If the editor feels particularly strongly on this point we would be happy to make changes, but would be concerned that data on overall missingness would then not be included. We have adjusted our limitations section as suggested, although this limitation was already included “Relatively high rates of missing data on participant ethnicity are in keeping with typical results taken from real-world primary care databases, but this missing data is an important limitation to interpretation of findings related to ethnicity, and capture of this data in future observational cohorts would be of clear benefit.”

While it is clearly an important aspect of research, particularly for equity issues, to capture data on ethnicity, the data set is a live electroinic medical registry formed of linked primary and secondary care data; as a research team we do not have the ability to influence the capturing of data.

Nonetheless we have included this in our future directions section – “High levels of missing data regarding patient ethnicity may limit understanding, and future focus on capturing this within electronic health records may be of benefit in redressing health inequalities.”

- Last cell contents should be in a footnote as only for age and medications. Please check the latter i.e. ? 49 medications per person.

The latter value is the total number of medications prescribed to an individual over the study period, including repeat prescriptions. The mean(sd) has been removed in line with the footnote.

Table 2

- Numbers are not clear. The total should be in the title.

This has been added

- The title should clearly differentiate these analyses from table 1.

The title of this table includes “presented by primary care utilisation tertiles”, whereas table 1 presents “Summary sociodemographic and clinical characteristics.” We hope this distinction is now clear, and as above we have altered both the methods and results sections to further aid clarity.

- You need an in-text reference.

We are not clear quite what the reviewer means by this. – please accept our apologies. Presumably they are suggesting we refer to Table 2 in the main text? This is already included in the methods and results.

- Ethnic proportions should not include missing data – see above.

The proportions of missing data are clearly included within this table, again to highlight to the reader the applicability to their own patient groups and aid in critical appraisal of generalisability. The proportions of participants with a coded ethnicity who are, for example, white can be calculated from the results we provide. We feel that providing less data and only presenting numbers for non-missing data would in fact hinder, rather than aid, transparency of findings.

- Check spacing.

This has been reviewed – thank you. We apologise for any persistent issues with spacing.

- Consider commas in numbers as they are hard to read when so large.

We reviewed journal style and found that few model papers we came across used commas. We are of course happy to include this if the editor or proof reader suggests.

Table 3

- Review layout of Table 3

This has been reviewed, but it is unclear exactly what changed the reviewer would recommend, if any. Our feeling is that the presentation in its current format is reasonably clear, but we are of course open to changing how this is presented if it were felt this would improve clarity.

- Shorten heading and avoid repetition with headings in the table. The description in title is a repetition of your methods so shorten and reword this.

We felt for table 3 it was important the reader was able to understand the information being presented as a “self-sufficient” entity, ie, they did not need to refer back to the methods to understand what data was being presented.

- Use a footnote for the CV events definition.

This has been amended

- Review footnote - the OR is the header row so you don't need to define that.

This has been removed

- For the footnote use symbols not 'model' for confounding variables.

This has been amended

- Review spacing and fonts.

We are not clear from this comment exactly which aspects of font or spacing should be altered in this table – we could not find any specific inconsistencies in font or errors in spacing. We are of course more than happy to amend as needed.

Figures 1 and 2

- Suggest you add Q1 – Q4 in titles i.e. ...by year quarters (Q1 – Q4). Description of quarters is in your text.

This has been amended as suggested

- Centre titles and A and B are usually written as 2a and 2b – check Journal format.

This has been amended as suggested

Discussion

Some restructuring is required. Separate paragraphs for increased utilisation and nonassociations with improved outcomes and link these comments to other literature.

We are a little unclear on exactly what the reviewer means by these comments. The “comparison to existing literature” section already contains separate paragraphs on increased utilisation, and nonassociation with improved outcomes; and these are compared to existing findings and reports in the literature. If the reviewer feels particularly strongly on this point, please could they expand a little further? We are of course happy to make any changes felt necessary.

- Move strengths and limitations to the end.

Altered as suggested

- Page 6 lines 45-49 – suggest this as your conclusion.

This has been moved to the conclusion, replacing a similar paragraph in this section

- Page 6 lines 55-57 - this is not clear. Are you referring to your results and state for which patients might benefit rather than 'selected' and 'unselected'.

This has been rewritten and restructured for clarity.

- Lines 59-60 is a repetition of your first paragraph – I suggest you link this into comparative studies.

Again we are a little unclear on what exactly the reviewer means by this comment. Line 59-60 is already within the "comparison to existing literature" section; how would they suggest further linking it in, if this is indeed needed?

- Page 7 lines 3-8 are not very clear and you are citing qualitative self-reporting from GPs and specialist nurses on the constraints and the effect they might have on patients at high risk of major CV events. What quantitative reports could you cite. This association between primary care consultations and outcomes has been identified in other countries and highlights the many challenges and limitations in managing and supporting people with type 2 diabetes and particularly those already at high risk of related complications including major CV events.

We would respectfully disagree with your reviewer that the value of citing qualitative literature in this field is limited. By this point in the manuscript we have already made several comparisons to the highest quality quantitative studies of which we are aware (Hobbs and Lay-Flurrie et al). We feel that adding a further quantitative comparison here offers less value than exploring, in the discussion, possible reasons underlying this association. Understanding why these associations may be present beyond the observations reported in quantitative cohort studies requires alternative methodological approaches, including qualitative work. Furthermore, the qualitative findings we report are not "self-reported" but from rigorously conducted semi-structured interviews performed in line with gold standards for qualitative methodology.

While we would be happy to remove this section, if the reviewer felt particularly strongly, we maintain that this qualitative perspective adds value and context in interpretation of our results.

- Your findings suggest that increasing consultations is not reducing major events and mortality. Do you have data on the proportion of patients with high HbA1c, hypertension, smoking and high LDL-C and prescription medications? What proportion of patients require further pharmaceutical interventions? You may be planning another paper with this information, but you could indicate this or recommend further analysis to identify gaps in CV risk management.

This is an excellent point highlighted by your reviewer. While these analyses would certainly be interesting, they are outside the scope of the current analysis, and the lead author (SH) is currently on a year of parental leave; it would not be possible to undertake further extensive analyses prior to resubmission. These suggestions could certainly form the basis of future research, as your reviewer rightly suggests. The strengths and limitations section has been adapted accordingly.

- Implications – first para is another summary of results. Can you comment on what is required to identify gaps in practice to reduce the risk profile in those at high risk of CV events? How often do these high risk patients have a CV risk assessment and review of medications and lifestyle?

This is an excellent point and the implications section has been updated accordingly

Grammar

- Abbreviations – check throughout i.e. UK, NHS, IMD. Either use CV or CVD and check as sometimes you have CVD then disease written afterwards.

We thank the reviewer for highlighting this inconsistency and have updated accordingly

- Use correct in-text references style and spacing for this Journal.

References have been formatted using the Mendeley "BMJ" style. This can be amended if required by the editorial team.

- Check and reference all data i.e. 80% in first para

We thank the reviewer for highlighting this omission, which has been corrected

- P. 4 lines 13-14 – check punctuation.

Thank you – the additional open bracket has been removed.

- Check Journals use of small headings. I strongly suggest you reduce the number of small headings and use full sentences and paragraph structure especially noticeable in the methods section.

We would be very willing to remove small headings, including in the methods section, if this improves readability. At present, we have elected to leave this in place to aid with cross-referencing with the STROBE checklist, and to allow the reader to quickly understand what variables were used and how in analyses.

- UK spelling needs to be consistent.

We have rechecked through the manuscript and removed any non-UK spellings we could identify

- Capitals when referring to Tables and Figures in the text.

This has been updated

- Check UK census and follow this where possible for ethnic groups i.e. 'white' is not the same as English, Irish and Welsh.

This is a useful point, but we do not have access to this data. The data provided by CHIA does not provide census-style ethnicity reports; we have presented the ethnic groupings as provided to us, ie "White." We would not wish to misrepresent the data in our report and have therefore used the term "White" throughout.

- P 5 lines 53-60 check parenthesis, spacing and sentence structure for mean and SD.

On reviewing it appears the spacing issues suggested are due to MS Word justification, as opposed to double spaced.

- Review journal lay out of tables and titles.

Tables and titles have been amended as suggested above

- Consider commas in large numbers for readability.

We have at present not added commas but would be happy to insert if the editorial team felt this necessary

- Page 5 line 47-49 join sentences re age and ethnicity.

Having joined these sentences, it became difficult to ensure clarity of meaning; the sentence was long and overly wordy. We have reverted back to the original structure but would be happy to adjust along specific suggestion lines if necessary.

Reviewer 2

This study is a longitudinal retrospective cohort study that explores the number of primary care visits for adults with type 2 diabetes during the time period April 2013 through April 2020. The findings demonstrate that higher utilization of primary care services resulted in a higher number of cardiovascular events and all-cause mortality. In Table 2, mean weight is presented as a variable, but it would be much more informative if the authors could include mean BMI and adjust findings for BMI. The discussion clearly suggests the limitations of the study (including the retrospective, observational nature of the study design) therefore, limiting inferences on causality and also, the limited racial/ethnic diversity of the population studied. While the authors acknowledge that their findings may be attributed to "reverse causality" (i.e., sicker patients needing more frequent care), they also suggest that their findings demonstrate that increasing frequency of services does not necessarily translate into improved outcomes. The call to conduct more extensive research on the quality of visits, rather than simple quantity is justified.

We thank the reviewer for these helpful comments. We can identify only one suggestion (to replace weight with BMI in table 2). We apologise for this error. As your reviewer highlights, subsequent analyses were already adjusted for BMI (not weight). We have amended this error and included BMI in the table now.

VERSION 2 – REVIEW

REVIEWER	Daly, B. University of Auckland
REVIEW RETURNED	23-Dec-2021

GENERAL COMMENTS	23/12/2021 Second Review for BMJ Open 'ID bmjopen-2021-054654' Titled 'Primary care service utilisation and outcomes in type 2 diabetes: A longitudinal cohort analysis' General feedback Authors appear to have made all the main changes requested. There are still a small number of edits to make. Mostly spacing before brackets, shortening some new sentences and abbreviations – outlined below. Editing required • Check in text reference spacing – see BMI Open publications. Stops and commas go before the reference-check all.• Correct all abbreviations i.e only write cardiovascular once then CVD and check all.• Check spacing around all brackets in text and tables.• Correct spacing in tables i.e right aligned.• Page 5 line 32 – write a short sentence and explain why patients were not involved – i.e. anonymous records...• Page 7 Line 45 - Spit into two sentences i.e. stop after diabetes and before 'For example...• Page 8 line 11-14 . long sentence spilt into 2 or 3.
--